# Effect of Rolling Process and Aging on the Microstructure and Properties of Cu-1.0Cr-0.1Zr Alloy

**DOI:** 10.3390/ma16041592

**Published:** 2023-02-14

**Authors:** Jun Zha, Yu Zhao, Yihui Qiao, Haohao Zou, Zeen Hua, Weiwei Zhu, Ying Han, Guoqing Zu, Xu Ran

**Affiliations:** 1Key Laboratory of Advanced Structural Materials, Ministry of Education, Changchun University of Technology, Changchun 130012, China; 2School of Materials Science and Engineering, Changchun University of Technology, Changchun 130012, China

**Keywords:** Cu-1.0Cr-0.1Zr alloy, cryogenic rolling, microstructure, mechanical properties, friction and wear

## Abstract

In order to study the effect of the rolling process and aging on the microstructure evolution and mechanical and tribological properties of the material, room-temperature rolling (RTR), cryogenic rolling (CR), and deep cryogenic treatment after rolling (RTR + DCT) experiments were carried out on a Cu-1.0Cr-0.1Zr alloy by a large plastic deformation process. Alloy plates were aged at 550 °C for 60 min. Different rolling processes and aging treatments have different effects on the microstructure and properties of alloy plates. The alloy plate is rolled and deformed, and the grains change from equiaxed to layered. Compared with RTR and RTR + DCT treatment, CR can promote the precipitation of the Cr phase and the degree of grain fragmentation is greater. After aging treatment, the Cu-Zr mesophase compounds in the microstructure increased, the alloys treated with CR and RTR + DCT appeared to be partially recrystallized, and the number of twins in the CR alloy plate was significantly more than that of RTR + DCT. The ultimate tensile strength of the alloy plate reached 553 MPa and the hardness reached 170 HV after cryogenic rolling with 90% deformation, which indicates that CR treatment can further improve the physical properties of the alloy plate. After aging at 550 °C for 60 min, the RTR 90% + DCT alloy plate has a tensile strength of 498 MPa and an elongation of 47.9%, which is three times that of the as-rolled alloy plate. From the research on the tribological properties of alloy plates, we learned that the main wear mechanisms in the wear forms of CR and RTR + DCT alloy plates are adhesive wear and abrasive wear. Adhesive wear is dominant in the early stage, while abrasive wear is the dominant mechanism in the later stage of wear. The friction coefficient of the CR 90% alloy plate in the TD direction is close to 0.55, and the wear rate is 2.9 × 10^−4^ mm^3^/Nm, indicating that the CR treatment further improves the wear resistance of the alloy plates.

## 1. Introduction

Cu-Cr-Zr is a typical precipitation-strengthened alloy. Due to its advantages, such as high strength, high electrical conductivity, and good tribological properties, it is currently the ideal Cu alloy for connectors in high-current and high-voltage applications [1,2,3,4,5]. However, it is difficult to achieve high strength and high ductility in Cu alloys [6]. In order to meet the increasing industrial demand for Cu-Cr-Zr alloys, the commonly used strengthening methods are solid solution, aging, strain hardening, and grain refinement [7]. Liu et al. [8] treated a Cu-0.55Cr-0.05Zr alloy with solution treatment at 1050 °C and then aging at 600 °C for 1 h. The final alloy had a hardness of 131 HV. Fu et al. [9] found that forming bulk Cr and Cu4Zr precipitates during aging can significantly improve the strength and electrical conductivity of Cu-Cr-Zr alloys. Meng et al. [10] adopted a Cu-0.4Cr-0.3Zr alloy aged at 450 °C for 3 h and then rolled at room temperature with 80% deformation to obtain a tensile strength of 568 MPa. Pang et al. [11] studied an 80% cold-rolled Cu-Cr-Zr alloy aged at 450 °C for 6 h with a hardness of 175 HV.

Deep cryogenic treatment (DCT) refers to the use of low-temperature liquid substances (such as liquid nitrogen) as refrigerants to process materials in an environment below −130 °C to improve material properties, which is an innovation of traditional heat treatment [12,13]. In the past decade, DCT has been an important method to enhance the properties of ferroalloys, amorphous alloys, and some non-ferrous alloys [14,15]. Klug et al. [16] believed that the factors affecting cryogenic treatment are usually cryogenic treatment time, temperature, cooling rate, and the location of the sample in the cryogenic device. Barylski et al. [17] studied the effect of DCT on the microstructure, wear, and mechanical properties of Mg-Y-Nd-Zr alloys and found that DCT accelerated the precipitation of solid solution atoms, thereby improving the strength of Mg-Y-Nd-Zr, reducing the formation of micro-cutting and deep scratches during the wear of Mg-Y-Nd-Zr alloys. The service life of the alloy was improved. DCT can significantly improve material performance [18]. Different rolling processes have different effects on material properties. D. Rahmatabadi et al. [19], in order to study the effect of post-annealing and ultrasonic vibration on the formability of multilayer Al5052/MgAZ31B composites, used two-pass cool roll bonding (CRB) technology to produce multilayer Al5052/MgAZ31B composites, and then annealed the materials to improve the interfacial bonding quality. They found that the cold rolling process can significantly improve the tensile strength and microhardness. Moslem Tayyebi et al. [20] processed Al/Cu/Mg multilayer composites by accumulating rolling (ARB) process followed by annealing at different temperatures and times. Studies have shown that the cumulative strain under various annealing conditions has minimal effect on the thickness of the intermetallic compound, and its effect is more to accelerate the nucleation of the compound. Furthermore, the heat treatment at different temperatures and times and the rolling times of ARB formed Al_3_Mg_2_ intermetallic compound layers with equiaxed columnar structures. According to previous studies, through proper deformation and aging, combined with cryogenic treatment, the alloy could have higher strength and wear resistance, which has a broad application prospect in today’s social development [21].

Therefore, this paper focuses on the effects of the rolling process and aging treatment on the microstructural, mechanical, and tribological properties of the Cu-1.0Cr-0.1Zr alloy and explores the performance strengthening mechanism, aiming to optimize the combination properties of the alloy.

## 2. Experimental Material and Procedures

In this experiment, 10 mm Cu-1.0Cr-0.1Zr alloy plates were used as the initial material, and the specific chemical composition is shown in Table 1. The plates were solution-treated (ST) at 960 °C for 60 min, rapidly water-cooled, and treated by RTR, CR, and RTR + DCT. In the process of cryogenic rolling and deformation, the method of combining cryogenic and cyclic rolling is adopted, the plates were placed in a liquid nitrogen environment for 10 min of cryogenic cooling in the gap between cyclic rolling. After the plates were treated by RTR, CR, and RTR + DCT, three groups of alloy plates with different rolling treatments were obtained, and each group of plates was subjected to true strains of 0.69, 1.20, and 2.30. The three groups of alloy plates subjected to different rolling deformations were aged at 550 °C for 60 min. The experimental flow is shown in Figure 1. Finally, the as-rolled and aged alloy plates were tested on the friction and wear tester at a sliding speed of 40 mm/s under the load of 10 N and 20 N.

The microstructure and morphology were observed by Leica Q550IW optical microscope and the JSM-5800 scanning electron microscope. The D/Max2500 X-ray diffractometer analyzed the phase composition of the alloy. The mechanical properties of the alloy plates were investigated by tensile and microhardness tests. Tensile specimens were prepared in the rolling direction (RD) according to the ASTME8/E8M-9 standard, and performed on a WDW-20 electronic universal testing machine at a strain rate of 1 × 10^−4^/s. All tests were performed at least twice, and a third test was performed if the difference between the first two samples exceeded 5%. All alloy plates were tested for microhardness using a JENUS instrument with a load of 50 g and a dwell time of 10 s. The MFT-5000 Rtec multifunctional friction and wear testing machine was used to conduct friction and wear tests on the RD and TD directions of the plates. The ZEISS Imager.2 m laser confocal microscope and scanning electron microscope were adopted to observe the worn surface morphology.

## 3. Results and Discussion

### 3.1. Microstructure and Phase Analysis

The microstructure of the Cu-1.0Cr-0.1Zr alloy under different processes is shown in Figure 2. After solution treatment, the grains are uniform and equiaxed with clear grain boundaries in Figure 2a. The black particles in the Cu matrix mainly gather at the grain boundaries. EDS analysis in Figure 2c shows that the composition of the black precipitates is primarily composed of Cu and Cr, and the mass percentage of Cr is as high as 85.71%. The crystal structure of the Cr phase in the copper matrix is mainly fcc. In the solid solution process, the solid solution transitions from a supersaturated solid solution to a metastable fcc ordered phase through a solute-rich region, and finally forms a bcc ordered phase [22]. In the alloy structure after aging treatment in Figure 2d, the Cr phase decreases and the Cu-Zr mesophase compound increases. EDS analysis shows that the spherical off-white precipitates in the crystal have a higher content of Cu and Zr, and the mass percentages of the two elements are about 36.89% and 54.35%, respectively. This phenomenon has also been reported before [23,24,25]. 

After rolling in Figure 3a–c, the grains were observed to be elongated and evolved into a layered structure. With the further increase in rolling deformation, more lamellar structures are formed, and the grain boundaries change from clear to fuzzy. Compared with the RTR-treated plate in Figure 3a, the black and white Cr phase and Cu-Zr mesophase precipitates are more uniformly distributed in the matrix after RTR + DCT treatment in Figure 3d, which indicates that the DCT treatment contributes to the dispersed Cu-Cr-Zr alloy [26]. The precipitation of small particles improves the mechanical properties of the alloy. In Figure 3e,f, the microstructure of the alloy changed after the aging of the alloy plates treated by different rolling processes. As the aging time increases, the microstructure of the alloy treated by RTR 90% + DCT and CR changes from a fibrillation to a recrystallization state. However, the degree of recrystallization of the CR-treated plate after aging is obviously stronger than that of RTR + DCT [27,28]. 

In order to further study the influence of rolling and aging on the microstructure, the X-ray diffraction spectra of alloy plates in different states were analyzed. In the alloy plates after rolling deformation, the (111) peak and (220) peak of Cu phase are weakened in Figure 4a, and such changes indicate that the grains are broken by rolling, the 90% rolling deformation is compared with the solid solution state, and the overall peak intensity is low and the peak width is small. It can be concluded that the alloy structure will recover under large deformation, and the grains will be elongated and broken to form equiaxed crystals. After CR treatment, the Cr (110) peak was enhanced to the highest level, and the Cu (111), (220), and (222) plane diffraction peak intensities began to weaken, because the grains were refined and the preferred orientation appeared during the CR treatment [29]. In Figure 4b, the peak of Zr (311) appears after aging, indicating that aging can effectively promote the precipitation of Cu-Zr mesophase compounds.

### 3.2. Mechanical Properties and Microhardness

The engineering stress–strain curves of the Cu-1.0Cr-0.1Zr alloy as-rolled and aged are shown in Figure 5. The as-rolled Cu alloy plate has the highest tensile strength but low elongation. Compared with RTR treatment, CR treatment can significantly improve the tensile strength of the alloy. Table 2 gives information on the strength and elasticity of each sample. The ultimate tensile strength (UTS) of CR 90% alloy plates increased from 484 MPa of RTR to 553 MPa, an increase of about 71 MPa, and the elongation is similar to that of the RTR alloy plates. After CR treatment, the microstructures of alloy plates are more refined, and the precipitates are randomly distributed in the matrix, which increases the dislocation density and thus enhances the strength of the alloy [30,31,32]. After aging treatment, the ultimate tensile strength (UTS) of the alloy plates was reduced, but the elongation was significantly improved compared with the as-rolled alloy plates. Among all alloy plates, the ultimate tensile strength (UTS) of the RTR 90% + DCT plates after aging is 485 MPa, which exceeds the tensile strength of the RTR plates. At the same time, the elongation rate reaches 47.9%, which is three times that of the as-rolled alloy plates. Therefore, the alloy plates after CR 90% aging have excellent performance, their tensile strength reaches 540 MPa, and the elongation reaches 38.1%.

Further, in order to understand the influence of the rolling process and aging on the hardness, the hardness of the Cu-1.0Cr-0.1Zr alloy plates in different states was tested, as shown in Figure 6. It can be seen from the figure that as the true strain increases, the hardness of the alloy plates also increases. Compared with RTR alloy plates, the hardness of the CR and RTR + DCT alloy plates increases significantly. CR alloy plates achieve the highest hardness of 170 HV when the true strain is 2.3. Since the alloy plates are subjected to strain hardening during the rolling deformation process, the trend of the influence of CR and RTR treatment on hardness is the same. The overall trend is rising, but with the increase in true strain, the hardness of CR 90% alloy plates has the highest degree of improvement compared with the other two groups of processes, which form the initial 103 HV to 170 HV. After aging treatment, the hardness curve shows a trend of increasing and then decreasing, which is the interaction of precipitation strengthening and strain hardening [33]. The alloy plates with a true strain of 0.63 by CR treatment have the highest hardness of 155 HV. Comparing the hardness of the as-rolled alloy plates, the hardness of the alloy plates with a high degree of true strain decreases most obviously after aging. Specifically, the hardness of CR alloy plates decreased by 45 HV. Low-strain alloy plates are less affected by strain hardening. After aging, the effect of precipitation strengthening is stronger than that of strain hardening, so the hardness value tends to increase. The alloy plates with a high strain degree before aging have an obvious strain-hardening effect, and the hardness shows an upward trend. During the aging process, the dislocation density gradually decreases, the strain hardening effect weakens, and the precipitation strengthening is not obvious, resulting in a decrease in the hardness of the alloy.

### 3.3. Fracture Morphology Analysis

CR and RTR + DCT processes significantly improve the properties of alloy plates. Combined with the stress–strain curve (Figure 5), the two treatment processes have different effects on the mechanical properties of the alloy before and after aging. In order to better understand the effect of CR and RTR + DCT treatment on the alloy plates, we observed the fracture after stretching (Figure 7). Tensile fractures treated by different rolling processes present different microscopic morphologies. The fracture morphology of the CR 90% alloy plates is shown in Figure 7a,b, and the fracture dimples of the alloy are not obvious. There are black precipitates in most of the dimples, and after EDS analysis the content of Cr in the precipitates is relatively high. Figure 7c,d shows the fracture morphology of the CR 90% alloy plates after aging treatment, the fracture toughness and the number of dimples has increased, and the black precipitates in the dimples have disappeared. Compared with the as-rolled alloy plates, there are a few white precipitates, and the dimple size has decreased to a certain extent. After aging treatment, many dimples appeared in the middle of the fracture of the alloy plates, while the surrounding area showed tearing edges and ridge-like patterns. During the fracture process, the brittle fracture of the surface layer transitioned to the interior of the alloy and transformed into the ductile fracture. The fracture morphology of RTR 90% alloy plates in Figure 7e,f is ridge-like, the local dimples are scattered and deep, and the proportion of nearby cleavage planes is high, which is a typical brittle fracture. The fracture morphology of the RTR 90% + DCT alloy plate is shown in Figure 7g,h. There are many dimples in the fracture, some of which are relatively large, and there are tear edges and ridge-like patterns, which are typical ductile–brittle mixed fractures. In the fracture morphology, the number of dimples decreases, the size of dimples becomes larger, cleavage fracture occurs, and the plasticity of the material is greatly reduced.

### 3.4. Friction and Wear

After analyzing the microstructure and mechanical properties of the alloy plates, CR can effectively improve the hardness. In general, the higher the hardness of the metal material, the smaller the friction coefficient [34]. In order to explore the effect of the rolling process on the tribological properties of the alloy plates, further friction and wear experiments were carried out in the RD and TD directions of the rolled plates. Figure 8 is the friction coefficient of the Cu-1.0Cr-0.1Zr alloy with 90% rolling deformation under two loading pressures. With the increase in wear time, the alloy friction coefficient fluctuates greatly in the early stage and tends to be stable later. The time-varying curves of the friction coefficient of the rolled plate with 90% deformation in the RD and TD directions can be seen in Figure 8a. With the increase in loading pressure, the friction coefficient increases slightly. In the RD direction, the friction coefficient of the RTR alloy plates is about 0.75, and that of the CR alloy plates is about 0.65. In the TD direction, the changing trend of the friction coefficient of the alloy plates is consistent, and the CR has a lower friction coefficient, only 0.55. After aging in Figure 8b, the friction coefficient curve changes, and the friction coefficient of the CR alloy plates in RD and TD increases significantly. Combined with the hardness before and after aging in Figure 5, CR can increase the hardness, thereby improving the wear resistance of the alloy plates.

According to the wear rate formula:(1)$$\delta =\frac{V}{\sum W}$$
(2)$$V=\int_{0}^{\omega }D\left(x\right)ldx,\;\sum W=\int_{0}^{L}\mu \left(x\right)Fdx=Fv\int_{0}^{T}\mu \left(t\right)dt,$$
(3)$$\delta =\frac{\int_{0}^{\omega }D\left(x\right)𝓁dx}{Fv\int_{0}^{T}\mu \left(t\right)dt}$$

*δ* is the wear rate, *V* is the wear volume, ∑*W* means cumulative friction work, 𝓁 is the wear scar length, ω is the wear scar width, *D*(*x*) is the function curve of wear scar width and depth, *μ* is the friction coefficient curve, *L* is the total sliding distance, *T* is the total sliding time, *v* is the sliding speed.

Figure 9 shows the wear rate of the Cu-1.0Cr-0.1Zr alloy under different loads in the TD direction before and after aging. It can be found from the figure that with the increase in the deformation amount, the wear rate of the alloy gradually decreases. Figure 9a is the wear rate of the alloy plates in the as-rolled state under a load of 10 N. It can be clearly seen that the alloy plates of CR have the lowest wear rate, and after cryogenic deformation, the wear rate drops from 4.1 × 10^−4^ mm^3^/Nm to 2.9 × 10^−4^ mm^3^/Nm, a drop of about 29%. Figure 10 is the wear scar cross-section curve of the Cu-1.0Cr-0.1Zr alloy in the TD direction. After CR treatment, the depth curve of the wear scar is relatively flat, while the curve of RTR fluctuates greatly. The wear rate of RTR + DCT after rolling is lower than that of RTR, and its wear rate drops from 9.0 × 10^−4^ mm^3^/Nm to 7.5 × 10^−4^ mm^3^/Nm, a drop of approximately 16.6%. According to the hardness curve in Figure 5, hardness is also an important aspect affecting wear, and the higher the surface hardness, the lower the wear rate. Figure 9b shows the wear rate of the alloy plates in the aging state under 10 N load pressure. After the load pressure increased, the wear rate of the three groups of alloy plates after aging decreased significantly.

Figure 11 is the friction surface morphology of the Cu-1.0Cr-0.1Zr alloy under different rolling treatments. It can be seen from the figure that the wear surface of the alloy is mainly composed of a peeling layer, an oxide layer, grooves, and wear debris [35]. Figure 11a,b show the wear scar morphology of the CR 90% plates along the TD direction under a load of 10 N. The peeled layers are small, the grooves are shallow, and a large amount of wear debris accumulates on the worn surface. However, looking at the wear scar morphology of the aged plates (RTR 90% + DCT) along the TD direction under a load of 10 N in Figure 11e,f, the crumbs are larger, and the degree of aggregation is slightly lower than that of CR 90%. The peeling layers on the worn surface are large, the grooves are wide and deep, and the wear debris on the surface is relatively large. The wear surface morphology of the alloy plates is the most direct reflection of the wear mechanism of the alloy [12]. In the wear morphology of the CR 90% and RTR 90% + DCT alloy plates in Figure 11a,b,e,f, the main wear mechanisms are adhesive wear and abrasive wear. According to the friction coefficient curve in Figure 8, the friction coefficients of the CR 90% and RTR 90% + DCT alloy plates fluctuate greatly in the early stage, which is due to the primary mechanism of adhesive wear. Since the hardness of the grinding material is much greater than that of the alloy, the alloy plates are cut during the grinding process, forming a groove. In the alloy plates of CR 90% during the friction process, the friction surfaces are subjected to normal load and shear stress parallel to the wear surface. Although the strength of CR 90% plates is high, the plasticity is low, and the local stress concentration causes the formation of cracks on the worn surface. Under the reciprocating friction of the abrasive material, the cracks are easy to expand, and the peeling layer falls off to form a large amount of wear debris. As time goes on, the wear debris becomes smaller. Therefore, in the later stage of wear, abrasive wear is the dominant mechanism.

On the contrary, due to the low strength of the RTR 90% + DCT plates but high elongation, the cutting effect of the grinding material on the alloy plates is obvious, the groove is deep, and the peeling layer is large. Hence, adhesive wear is the dominant mechanism during the wear period. From the 3D-Profilometry of the CR 90% and RTR 90% + DCT plates in Figure 11c,d, it can also be seen that the grooves of the RTR 90% + DCT plates are deeper. The friction curves of CR 90% and RTR 90% + DCT plates in Figure 8 have a downward trend in the later period. After scanning the surface of the plates after the friction and wear test, it was found that the mixed layer contained copper, oxygen, aluminum, silicon, and other elements. The appearance of the oxygen element shows that in the later stage of the test, the action of the grinding material caused the oxygen in the atmosphere to contact the friction surface, forming a smooth and dense oxide film, which greatly improved the wear resistance of the material.

## 4. Conclusions

In this paper, the effect of different rolling processes and aging on the structure and properties of the Cu-1.0Cr-0.1Zr alloy plates were studied, and the microstructure, mechanical properties, and tribological properties of the rolled and aged alloy plates were analyzed. The main conclusions are as follows:After different rolling treatments, the alloy plate transforms from equiaxed to layered when the crystal grains are in a solid solution state. Compared with the RTR and RTR + DCT treatments, the CR treatment resulted in a higher degree of grain fragmentation and promoted the precipitation of Cr phase. After aging treatment, the Cu-Zr mesophase compounds in the microstructure increased, and the alloys treated by CR and RTR + DCT showed partial recrystallization, and the number of twins in the CR alloy plate was more than that of RTR + DCT.Different rolling processes have different effects on the properties of the alloy plates. RTR can improve the mechanical properties and hardness of the plates, but the tensile strength of the alloy plates after CR treatment reaches 553 MPa, which is 71 MPa higher than that of the RTR alloy plates. After aging at 550 °C for 60 min, the tensile strength of the RTR + DCT and CR alloy plates did not increase significantly, but the elongation increased significantly. The elongation of RTR + DCT reached 47.3%, which was three times that of the rolled state.The hardness increases with the increase in the deformation. In particular, the hardness of the CR 90% alloy plates is the highest, reaching 170 HV. The alloy plates with a high strain degree before aging have an obvious strain-hardening effect, and the hardness shows an upward trend. The hardness of the alloy plates treated with CR is further improved under the action of precipitates. During the aging process, the dislocation density gradually decreases, the strain-hardening effect weakens, and the precipitation strengthening of the CR and RTR + DCT plates is not obvious, resulting in a more obvious decrease in alloy hardnessThe main wear mechanisms in the wear patterns of the CR and RTR + DCT alloy plates are adhesive wear and abrasive wear. The coefficient of friction fluctuates greatly in the early stages, which is due to the dominant mechanism of adhesive wear. In contrast, in the later stages of wear, abrasive wear is the dominant mechanism. The CR 90% alloy plates obtain the lowest friction coefficient in the TD direction, reaching 0.55, the wear rate of the alloy plates is 2.9 × 10^−4^ mm^3^/Nm, and indicates an obvious advantage compared to the commercial products.

## Figures and Tables

**Figure 1 materials-16-01592-f001:**
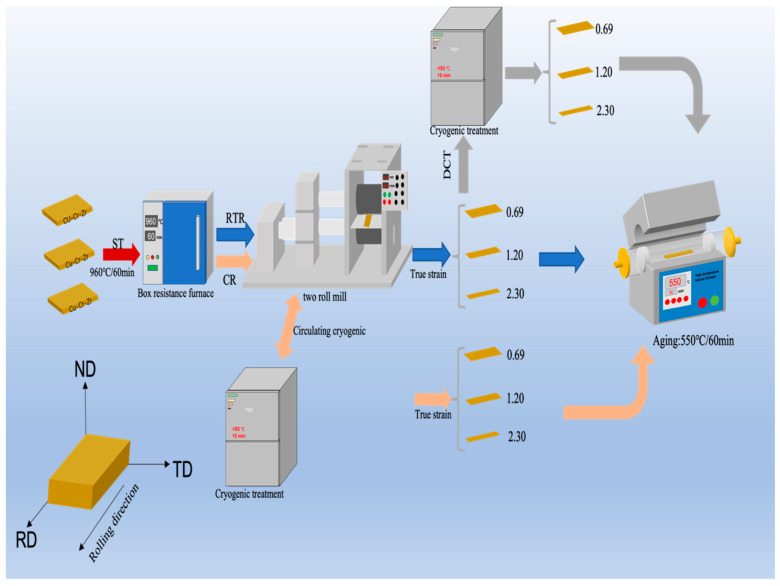
Experimental flow chart of the Cu-1.0Cr-0.1Zr alloy.

**Figure 2 materials-16-01592-f002:**
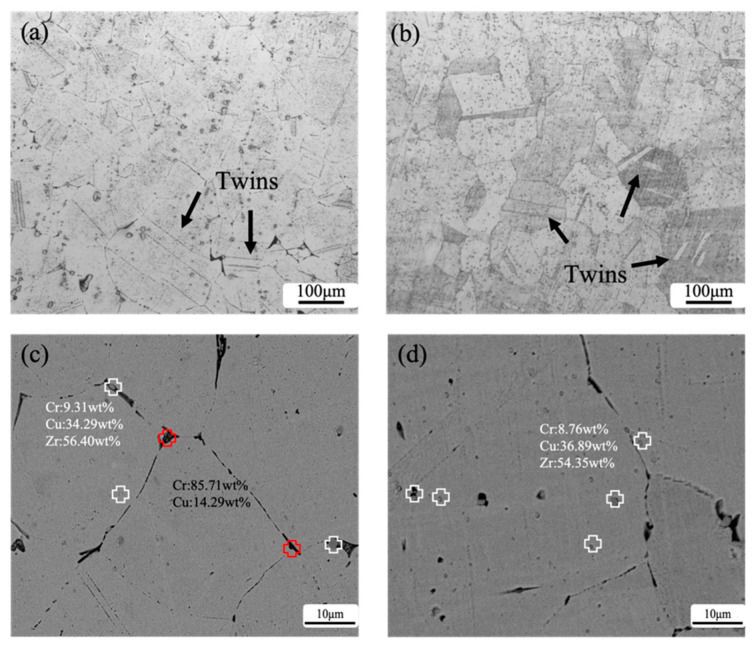
Microstructure of Cu-1.0Cr-0.1Zr alloys under different treatment conditions: (**a**) ST; (**b**) aging after ST; (**c**) SEM of ST; (**d**) SEM of aging.

**Figure 3 materials-16-01592-f003:**
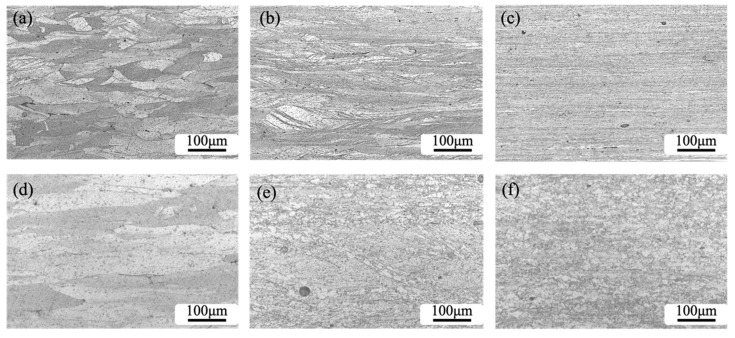
OM images of Cu-1.0Cr-0.1Zr alloys under different treatment conditions: (**a**–**c**) RTR 50%, RTR 70% and RTR 90%; (**d**) RTR 50% + DCT; (**e**) RTR 90% + DCT after aging; (**f**) CR 90% + DCT after aging.

**Figure 4 materials-16-01592-f004:**
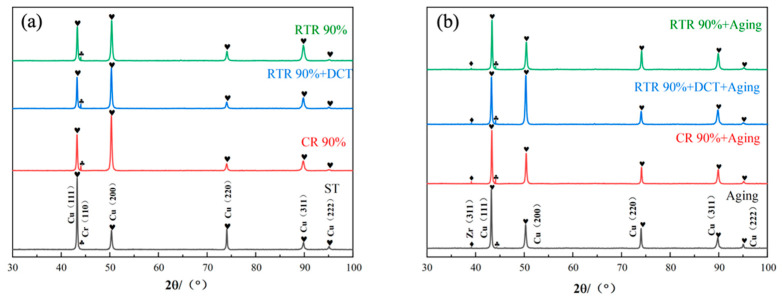
XRD patterns of Cu-1.0Cr-0.1Zr alloy at different states: (**a**) ST and as-rolled Cu-1.0Cr-0.1Zr alloy; (**b**) aging Cu-1.0Cr-0.1Zr alloy.

**Figure 5 materials-16-01592-f005:**
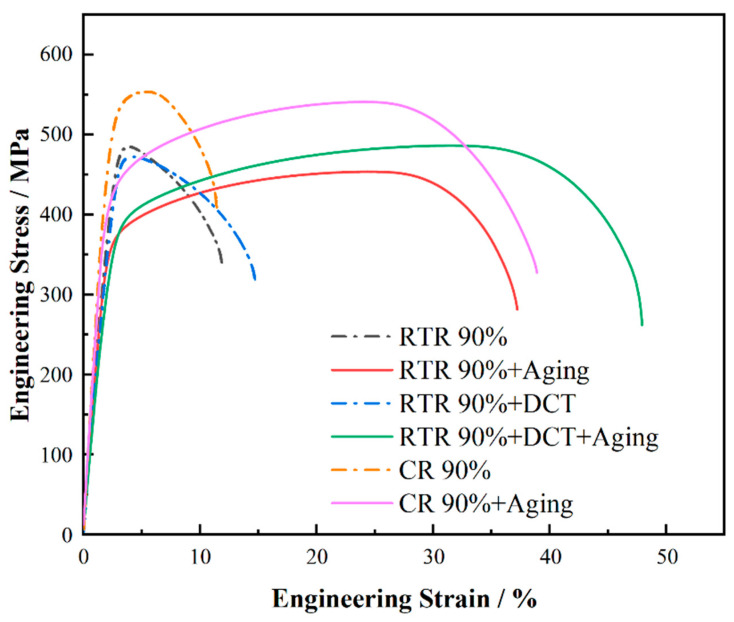
Engineering stress–strain curves of the Cu-1.0Cr-0.1Zr alloy after rolling and aging treatment.

**Figure 6 materials-16-01592-f006:**
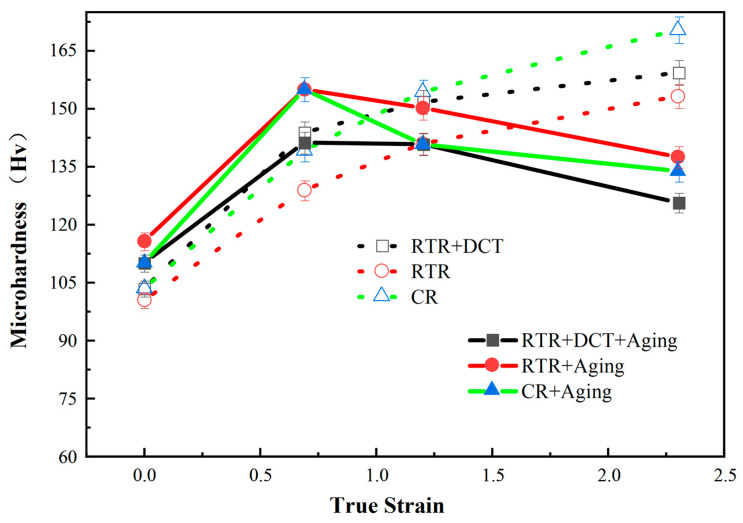
Hardness contrast curves of rolled Cu-1.0Cr-0.1Zr alloys at different states.

**Figure 7 materials-16-01592-f007:**
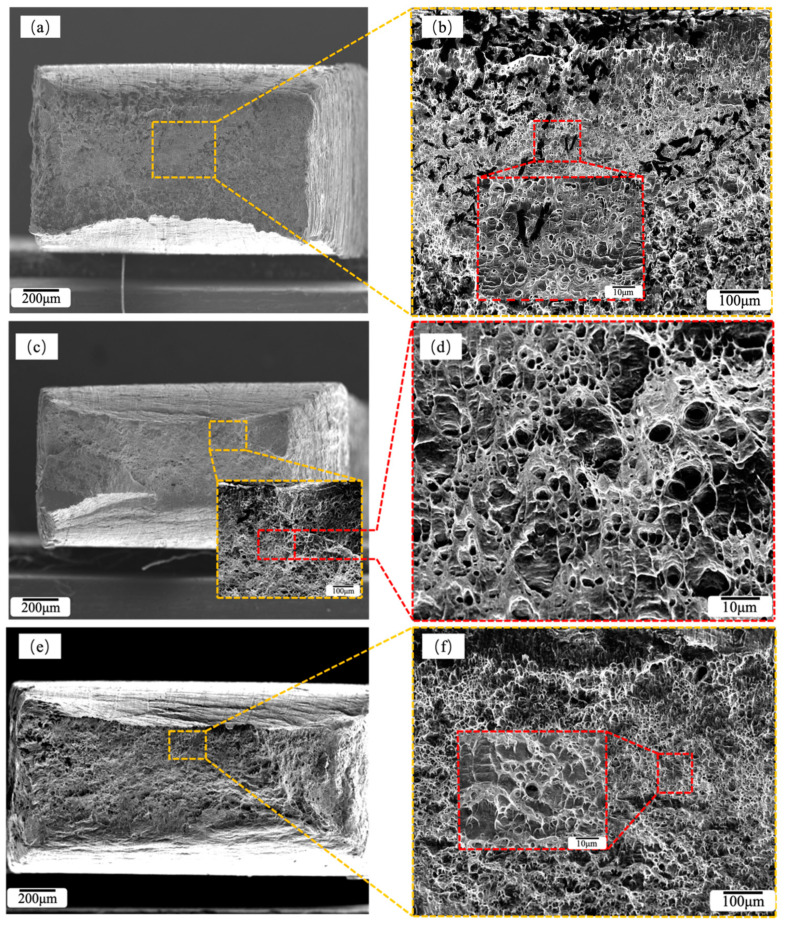
Fracture morphologies of Cu-1.0Cr-0.1Zr tensile specimen: (**a**,**b**) CR 90%; (**c**,**d**) CR 90% + Aging; (**e**,**f**) RTR 90%; (**g**,**h**) RTR 90% + DCT.

**Figure 8 materials-16-01592-f008:**
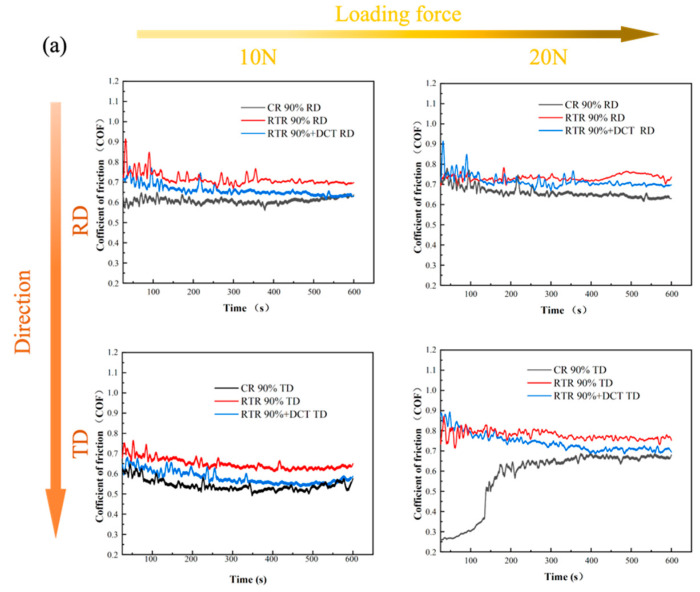
Friction coefficients of 90% rolled deformed Cu-1.0Cr-0.1Zr alloy in different rolling directions under two loading pressures: (**a**) as-rolled, (**b**) aging.

**Figure 9 materials-16-01592-f009:**
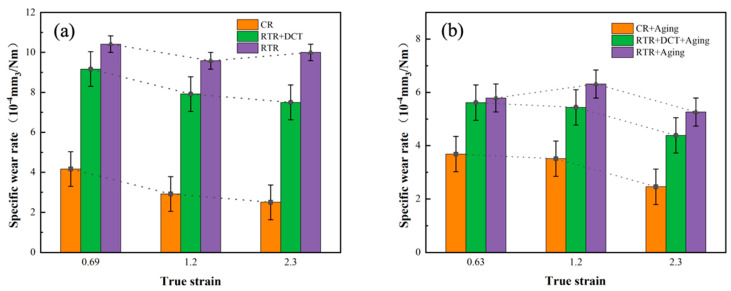
Wear rate of Cu-1.0Cr-0.1Zr alloy in transverse direction under 10 N load: (**a**) as-rolled, (**b**) aging.

**Figure 10 materials-16-01592-f010:**
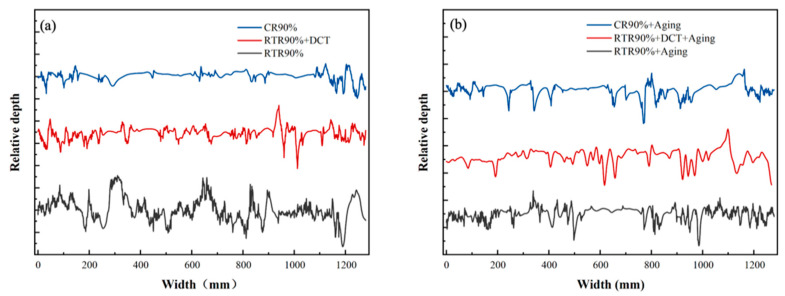
Cross-section curves of Cu-1.0Cr-0.1Zr alloy in transverse directions under 10 N load: (**a**) as-rolled, (**b**) aging.

**Figure 11 materials-16-01592-f011:**
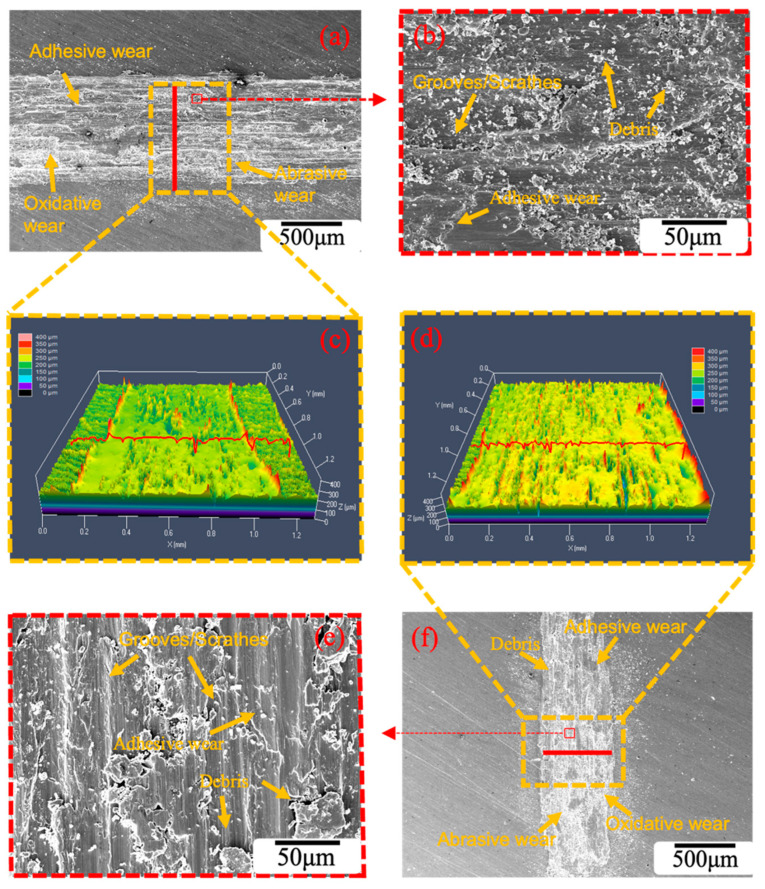
Morphology of the worn surface under different rolling treatments: (**a**,**b**) CR 90% under 10 N RD; (**c**) 3D-Profilometry of CR 90% as-rolled; (**d**) 3D-Profilometry of RTR 90% + DCT aging; (**e**,**f**) RTR 90% + DCT under 10 N RD.

**Table 1 materials-16-01592-t001:** Chemical composition of the Cu-Cr-Zr alloy.

Element	Cr	Zr	Si	Mg	Al	Cu
Content (wt.%)	1.0	0.1	0.05	0.25	0.25	balance

**Table 2 materials-16-01592-t002:** The variations in the ultimate tensile test and elongation.

Sample	$\mathrm{UTS}\;(\bar{X}\pm \;\sigma )$/MPa	$\mathrm{EL}\;(\bar{X}\pm \;\sigma )$/%
RTR 90%	484 ± 5.5	11.8 ± 0.8
RTR 90% + Aging	453 ± 5.3	37.2 ± 1.5
RTR 90% + DCT	472 ± 5.4	14.7 ± 0.9
RTR 90% + DCT + Aging	485 ± 5.5	47.9 ± 1.7
CR 90%	553 ± 5.6	11.3 ± 0.8
CR 90% + Aging	540 ± 5.8	38.1 ± 1.5

## Data Availability

Not applicable.

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
