# Peer review of "Effect of Rolling Process and Aging on the Microstructure and Properties of Cu-1.0Cr-0.1Zr Alloy"

_materials, 2023, doi:10.3390/ma16041592_

Round 1

Reviewer 1 Report

In this study, various rolling processes (room temperature rolling, cryogenic rolling, and deep cryogenic treatment after rolling) were applied to a Cu-Cr-Zr alloy, and the effects of these rolling processes on the microstructure evolution and mechanical and tribological properties of the material were investigated systematically and in detail. The results showed, in particular, the effectiveness of cryogenic treatments in improving the properties. This paper contains important results that deserve publication. It is well-written, in which the results are clearly presented. Therefore, the referee recommends the paper to be accepted for publication in Materials.

Error:

page 10, line 10

‘After aging in Fig. 8(b), …’

The same sentence is repeated.

Author Response

Dear Editor and Reviewers,

We would like to thank the reviewers for carefully reading our manuscript. We appreciate the comments and suggestions. In the following, we include a point-by-point response to the comments from each reviewer. In the revised manuscript, all the changes have been highlighted in red.

Point 1: Error:page 10, line 10‘After aging in Fig. 8(b), …’The same sentence is repeated.

Response 1: Thanks for pointing out our mistake, we have removed the duplicate sentence.

Reviewer 2 Report

I recommend the introduction of a sentence in the abstract that presents the conclusion of the study carried out.

In table 1, instead of “etc.” it is recommended to write “balance”.

In chapter 2, it is recommended to indicate the tensile test conditions as well as the hardness test method and equipment. Also here it must be specified what RD and TD represent.

In subsection 3.2, reference must be made to Figure 5 and not to Figure 4.

Figure 8 must be enlarged because the notations in the figure are not clear.

Author Response

We would like to thank the reviewers for carefully reading our manuscript.We appreciate the comments and suggestions. In the following, we include a point-by-point response to the comments from each reviewer. In the revised manuscript, all the changes have been highlighted in red.

Point 1: I recommend the introduction of a sentence in the abstract that presents the conclusion of the study carried out.

Response 1: Thanks for your suggestion, we have revised the abstract section. (in red)

Point 2: In table 1, instead of “etc.” it is recommended to write “balance”.

Response 2: Thanks for your advice. We have replaced 'etc' in Table 1 with your suggested words. (in red)

Point 3: In chapter 2, it is recommended to indicate the tensile test conditions as well as the hardness test method and equipment. Also here it must be specified what RD and TD represent.

Response 3:Thank you. We have added tensile test conditions and hardness test methods and equipment. In addition, we also elaborated the meaning of RD and TD in the paper. RD means rolling direction, TD means transverse direction (in red)

Point 4: In subsection 3.2, reference must be made to Figure 5 and not to Figure 4.

Response 4: Thank you for pointing out our mistake, which was caused by carelessness. We have modified it.(in red)

Point 5: Figure 8 must be enlarged because the notations in the figure are not clear.

Response 5: Thanks for your suggestion, we have modified Figure 8. (in red)

Reviewer 3 Report

Figures 2,3 and 8 need to be improved

Basic equations related to true stress-strain may be removed. 

Conclusions need to be rewritten indicating all the important findings

Objectives should be clearly written

Author Response

We would like to thank the reviewers for carefully reading our manuscript.We appreciate the comments and suggestions. In the following, we include a point-by-point response to the comments from each reviewer. In the revised manuscript, all the changes have been highlighted in red.

Point 1: Figures 2,3 and 8 need to be improved.

Response 1: Thanks for your suggestion, we have modified Figure 2,3 and 8. (in red)

Point 2: Basic equations related to true stress-strain may be removed. 

Response 2: Thanks for your advice. We have removed the basic equations related to the true stress-strain, which makes the presentation of the paper more natural and smooth.

Point 3: Conclusions need to be rewritten indicating all the important findings.Objectives should be clearly writte.

Response 3: Thank you for pointing out the problems with the conclusions of the paper. We have revised the conclusions to emphasize the effects of cryogenic rolling and aging on the properties of alloy plates. (in red)

Reviewer 4 Report

In this manuscript, very comprehensive research has been presented to investigate the effect of rolling temperature, aging after rolling, and the amount of strain during rolling on copper alloy, and the mechanical, microstructural, wear, and fracture properties have been investigated. Despite the achievements of this research, the abstract and conclusion need major corrections. Also, in order to improve the quality of the paper, the following items should be considered.

In the title, the effect of temperature and rolling conditions, as well as subsequent aging, is not presented, which is somehow the main goal of the research. The title should be corrected.

The introduction is written very superficially and needs major revisions. It is suggested to use the following sources to deepen the topics of rolling temperature effect and heat treatment.

https://doi.org/10.1016/j.actamat.2022.118248

https://doi.org/10.1016/j.jmrt.2021.06.032

https://doi.org/10.1080/02670836.2020.1867784

Equations 1 and 2 as well as the explanation of the relevant section can be deleted. It is enough to add the initial thickness and the thickness at the end of each rolling cycle.

Strains are applied in how many rolling passes? What is the reason for choosing these three?

The research method section and Figure 1 are confusing. Which samples are the three plates that have been aged?

The research method section and Figure 1 are confusing. Which samples are the three sheets that have been aged? Three types of rolling conditions are presented in the abstract if only DCT and RTR conditions are presented in Figure 1? CR is missing. Also, what is the difference between CR and DCT? Storage time in liquid nitrogen? Make this part clear.

Section 3-1 is more like a report of results and should be accompanied by more analysis and discussion. You can use the recommended resources. (Effects of minor Nd and Er additions on the precipitation evolution and dynamic recrystallization behavior of Mg–6.0Zn–0.5Mn alloy and Effect of heat input on interfacial characterization of the butter joint of hot-rolling CP-Ti/Q235 bimetallic sheets by Laser + CMT)

The first and second paragraphs of Section 3-1 are confusing (before and after Figure 2). First, it refers to figure 2, then figure 3, and back to figure 2 again.

What does SHT stand for? Does it mean ST?

The quality of images 3 is very low and should be improved.

It is better to divide these images into two or three images and present only the effect of strain in one and the effect of the rolling temperature in the other.

Also, for comparison, all images should have the same magnification.

What is the reason for the intensity and weakening of peaks after aging?

Section 3.2: What does after 90% rolling mean? The same terminology should be used throughout the article. (The true strain should be mentioned). This phrase should also be corrected RTR90%+DCT and CR90%.

In this paragraph, the term ultimate tensile strength is repeated three times along with its abbreviation (UTS).

It is suggested to summarize UTS and elongation values in one chart for better comparison.

Author Response

We would like to thank the reviewers for carefully reading our manuscript.We appreciate the comments and suggestions. In the following, we include a point-by-point response to the comments from each reviewer. In the revised manuscript, all the changes have been highlighted in red.

Point 1: In this manuscript, very comprehensive research has been presented to investigate the effect of rolling temperature, aging after rolling, and the amount of strain during rolling on copper alloy, and the mechanical, microstructural, wear, and fracture properties have been investigated. Despite the achievements of this research, the abstract and conclusion need major corrections. Also, in order to improve the quality of the paper, the following items should be considered.

Response 1: Thank you for pointing out the problems in the abstract and conclusion of the paper, we have revised the abstract and conclusion to emphasize the effect of rolling process and aging on the properties of alloy plates. (in red)

Point 2: In the title, the effect of temperature and rolling conditions, as well as subsequent aging, is not presented, which is somehow the main goal of the research. The title should be corrected.

Response 2: As you said, the title of the previous manuscript did not fully reflect the content of the paper. Thanks for your suggestion, we have revised the title. (in red)

Point 3: The introduction is written very superficially and needs major revisions. It is suggested to use the following sources to deepen the topics of rolling temperature effect and heat treatment.

https://doi.org/10.1016/j.actamat.2022.118248、https://doi.org/10.1016/j.jmrt.2021.06.032、https://doi.org/10.1080/02670836.2020.1867784

Response 3: Thank you for your guidance and providing relevant resources. We have carefully read and learned the relevant knowledge points, we have properly cited the resources you recommended, and modified the introduction. (in red)

Point 4: Equations 1 and 2 as well as the explanation of the relevant section can be deleted. It is enough to add the initial thickness and the thickness at the end of each rolling cycle.

Response 4: Thanks for your advice. We have removed the basic equations related to the true stress-strain, which makes the presentation of the paper more natural and smooth.

Point 5: Strains are applied in how many rolling passes? What is the reason for choosing these three?

Response 5: Different rolling mills have different parameter settings in the rolling process. We have done a lot of experiments. During the rolling process, the thickness of the plate is reduced by 10% each time. The possibility of cracking is greatly reduced. For the true strains of 0.69, 1.20, and 2.30, the rolling passes are 5, 7, and 9, respectively. In addition, the reason for choosing these three groups of strains is that while studying these three groups of strains, we also experimented with samples of other strains, but they are not as representative as these three groups in terms of tissue structure and performance, and it is easier to distinguish them. (in red)

Point 6: The research method section and Figure 1 are confusing. Which samples are the three plates that have been aged?

Response 6: We have revised the study section and Figure 1. After the plates were treated by RTR, CR and RTR+DCT, three groups of alloy plates with different rolling treatments were obtained, and the true strains of each group of plates were 0.69, 1.20 and 2.30, respectively. Each group of alloy plates subjected to different rolling processes was aged at 550°C for 60 minutes.

Point 7: Three types of rolling conditions are presented in the abstract if only DCT and RTR conditions are presented in Figure 1? CR is missing. Also, what is the difference between CR and DCT? Storage time in liquid nitrogen? Make this part clear. Section 3-1 is more like a report of results and should be accompanied by more analysis and discussion. You can use the recommended resources. (Effects of minor Nd and Er additions on the precipitation evolution and dynamic recrystallization behavior of Mg–6.0Zn–0.5Mn alloy and Effect of heat input on interfacial characterization of the butter joint of hot-rolling CP-Ti/Q235 bimetallic sheets by Laser + CMT)

Response 7: Thank you for expressing your confusion. First, RTR, CR and RTR+DCT are included in Figure 1. We use arrows of different colors to represent different technological processes. Secondly, DCT and CR are two different processing techniques. DCT refers to placing materials in a low-temperature environment, and mainly studies the influence of temperature and time on their properties. In the paper, RTR means room temperature rolling, and RTR+DCT means that after the material has been rolled at room temperature, the material is then subjected to low temperature treatment. CR is a kind of ultra-low temperature rolling that applies DCT technology. It performs DCT on the material before and after each pass of rolling to keep the material passing through the rolling mill in an ultra-low temperature state. Finally, in response to the questions you raised about section 3-1, we carefully read the relevant resources you provided, and revised the paper to make the analysis and discussion more specific and accurate. (in red)

Point 8: The first and second paragraphs of Section 3-1 are confusing (before and after Figure 2). First, it refers to figure 2, then figure 3, and back to figure 2 again.

Response 8: Thank you for pointing out our deficiencies, we have revised section 3-1.(in red)

Point 9: What does SHT stand for? Does it mean ST?

Response 9: Thanks for pointing out our mistake. This is a mistake in our writing, SHT refers to ST, we have corrected the wrong expression in the paper. (in red)

Point 10: The quality of images 3 is very low and should be improved. It is better to divide these images into two or three images and present only the effect of strain in one and the effect of the rolling temperature in the other.Also, for comparison, all images should have the same magnification.

Response 10: In response to your suggestions, we have modified Figure 3, thank you. (in red)

Point 11: What is the reason for the intensity and weakening of peaks after aging?

Response 11: Compared with the initial solid solution alloy plate, the Cu (111) peak of the as-rolled alloy plate is weakened and the (200) peak is enhanced, which indicates that the grain orientation has changed in the deformed structure. After aging, the (111) peak weakens and the other peaks intensify. This indicates that the lower deformation temperature changes the microstructure of the aged CuCrZr alloy. With the increase of the deformation amount, the (111) texture gradually weakens and new other orientations are formed. According to the relevant literature[1-4], the texture of the room temperature rolled sample is a strong Copper texture and Goss, Brass and S texture with weakened strength in turn, while the deep cold rolled sample is mainly a strong Goss texture and a relatively weak Brass texture.  Compared with solid solution Cu Cr Zr alloy, the texture transformation of aged Cu Cr Zr alloy and solid solution Cu Cr Zr alloy under cryogenic rolling conditions is similar, but not the same in room temperature rolling.

[1] T. Konkova, S. Mironov, A. Korznikov, G. Korznikova, M.M. Myshlyaev, S.L. Semiatin, Grain growth during annealing of cryogenically-rolled Cu–30Zn brass, J. Alloy. Compd., 666 (2016) 170-177.

[2] R. Kumar, S.M. Dasharath, P.C. Kang, C.C. Koch, S. Mula, Enhancement of mechanical properties of low stacking fault energy brass processed by cryorolling followed by short-annealing, Mater. Des., 67 (2015) 637-643.

[3] K. Abib, J.A.M. Balanos, B. Alili, D. Bradai, On the microstructure and texture of Cu-Cr-Zr alloy after severe plastic deformation by ECAP, Mater. Charact., 112 (2016) 252-258.

[4] J.J. Sidor, L.A.I. Kestens, Analytical description of rolling textures in face-centred-cubic metals, Scripta Mater., 68 (2013) 273-276.

Point 12: Section 3.2: What does after 90% rolling mean? The same terminology should be used throughout the article. (The true strain should be mentioned). This phrase should also be corrected RTR90%+DCT and CR90%.

Response 12: This is a typo in our language, thank you for pointing it out. 90% rolling refers to the rolling deformation with a true strain of 2.3, we have modified part 3-2. (in red)

Point 13: In this paragraph, the term ultimate tensile strength is repeated three times along with its abbreviation (UTS). It is suggested to summarize UTS and elongation values in one chart for better comparison.

Response 14: Thanks for the suggestion, we have added a table on UST and elongation.

Round 2

Reviewer 3 Report

The corrections are incorporated as specified.

Author Response

Dear Editors and Reviewers,
We would like to thank you for your efforts to improve the quality of this manuscript. Thank you for your valuable suggestions, which are very helpful to improve the quality of our papers, thank you.

Reviewer 4 Report

Some of the previous comments have not been answered well. Especially comment 7. In addition, in figures 7 and 11, the scale bar is illegible and should be corrected. The error bar should be added to Figure 6 and Table 2.

Author Response

Response to Reviewer 4 Comments

We would like to thank you for your efforts to improve the quality of this manuscript. We have made careful revisions based on the reviewer’s comments. In the revised manuscript, all the changes have been highlighted in red.

Point 1: Some of the previous comments have not been answered well. Especially comment 7.

Response 1: Thank you for your careful review of the revised version. We have carefully analyzed each suggestion and made improvements. For some of the previous comments, we have double checked and made some changes.

For comment 7, the three processes RTR, CR and RTR+DCT have been re-expressed in Fig. 1. DCT and CR are two different processing techniques. DCT refers to placing the material in a low temperature environment. Here we put the alloy plate in liquid nitrogen for 10 minutes. CR is a kind of ultra-low temperature rolling using DCT technology. It performs DCT on the material before and after each rolling pass, and the cryogenic time of each pass is 10 minutes to keep the material passing through the rolling mill at an ultra-low temperature. Combined with the revised Figure 1, the difference between the two groups of processes can be more intuitively understood.

Thank you for recommending relevant resources to us. We have carefully studied these two documents and cited them in the paper.

Point 2: In addition, in figures 7 and 11, the scale bar is illegible and should be corrected.

Response 2: Thanks for your suggestion, we have modified the scale bars of Figure 7 and 11.

Point 3: The error bar should be added to Figure 6 and Table 2.

Response 3: Thanks for your guidance, we have revised Figure 6 and Table 2 to improve the quality of the charts.
